# Functional Exercise Versus Specific Pelvic Floor Exercise: Observational Pilot Study in Female University Students

**DOI:** 10.3390/healthcare11040561

**Published:** 2023-02-14

**Authors:** Esther Díaz-Mohedo, Itxaso Odriozola Aguirre, Elena Molina García, Miguel Angel Infantes-Rosales, Fidel Hita-Contreras

**Affiliations:** 1Department of Physiotherapy, University of Málaga, 29016 Málaga, Spain; 2Freelance Physiotherapist, 31000 Pamplona, Spain; 3Freelance Physiotherapist, 29000 Málaga, Spain; 4Department of Physiotherapy, University of Jaén, 23071 Jaén, Spain

**Keywords:** pelvic floor, functional exercise, Kegel exercise, electromyography, women, push-up

## Abstract

Objectives: To evaluate the electromyographic (EMG) activity of the pelvic floor musculature (PFM) that takes place when performing the functional movement screen (FMS) exercise, comparing it with the activation in the maximum voluntary contraction of PFM in the supine position (MVC-SP) and standing (MVC-ST). Material and Methods: A descriptive, observational study conducted in two phases. In the first study phase, the baseline EMG activity of PFM was measured in the supine position and standing during MVC-SP and MVC-ST and during the execution of the seven exercises that make up the FMS. In the second phase of the study, the baseline EMG activity of PFM was measured in the supine position and standing during MVC-SP and MVC-ST and during the FMS exercise that produced the most EMG in the pilot phase: trunk stability push-up (PU). ANOVA, Friedman’s and Pearson’s tests were used. Results: All FMS exercises performed in the pilot phase showed a value below 100% maximum voluntary contraction (MVC) except PU, which presented an average value of 101.3 μv (SD = 54.5): 112% MVC (SD = 37.6). In the second phase of the study, it was observed that there were no significant differences (*p* = 0.087) between the three exercises performed: MVC-SP, MVC-ST and PU (39.2 μv (SD = 10.4), 37.5 μv (SD = 10.4) and 40.7 μv (SD = 10.2), respectively). Conclusions: There is no evidence of the existence of significant differences in EMG activation in PFM among the three exercises analysed: MVC-SP, MVC-ST and PU. The results show better EMG values in the functional exercise of PU.

## 1. Introduction

Urinary incontinence is very common in women of all ages; its prevalence in Europe and America varies between 5% and 42% [1,2]. Currently, the non-pharmacological and non-surgical treatment of urinary incontinence includes PFM training as first-line therapy (recommendation level A) [3,4], demonstrating its short-term efficacy [5,6,7,8,9,10]. However, the long-term efficacy of these exercises largely depends on adherence to the treatment [2]. Different systems have been used, such as electromyographic biofeedback [11,12] and mobile applications, with the aim of improving this aspect, showing a good but insufficient potential result [13]. Thus, it is still necessary to propose new therapeutic strategies that improve treatment adherence.

Nowadays, due to changes in lifestyle, so-called functional exercise or training is gaining special relevance due to its numerous benefits, and it has attracted a considerable part of the population to practice this type of training [14,15,16] in sports centres, which includes deep squats and trunk stability push-ups, among many others. These exercises are gathered in the FMS, which is a popular set of tests that evaluate the functionality and quality of movement; it consists of seven basic mobility tests that require a balance between mobility and stability (including motor control) [17]. Its inter- and intra-observer reliability has been demonstrated, and it is used to explore the functional asymmetries of the locomotor system and stability deficits [18,19,20]. In some cases, FMS exercises have been combined with diagnostic techniques, such as EMG, to identify the degree of activation of a specific muscle group during the execution of a certain movement [21]. 

The pelvic floor in women may be the only area of the body where the positive effect of physical activity and exercise has been questioned: exercising women have three times the risk of experiencing urinary incontinence [22,23]. In a recent review [24], Bø brought to light the existing controversy, describing two opposing and possible hypotheses on the effect of physical activity on the pelvic floor: (a) general exercise training strengthens the pelvic floor: exercise could lead to a co-contraction of the PFM, creating an indirect training effect, and (b) general exercise training overloads, stretches and weakens the pelvic floor: physical activity increases intra-abdominal pressure (IAP) and the PFM is not able to activate and co-contract quickly or strongly enough to counteract this increased pressure. 

Regarding Theory (b), two exercise modalities may increase IAP to a greater extent than others and, thus, possibly affect the pelvic floor and contribute to the incidence, progression or recurrence of pelvic floor disorders [25]: strength training and high-impact activities. 

It is important to consider FMS exercise as a strength exercise; understanding whether FMS exercise might predispose to, or prevent, dysfunction of the pelvic floor and, thus, these conditions is important. The aim of this exploratory study is to evaluate, through EMG, the degree of activation of PFM that takes place when performing the PU exercise included in FMS in nulliparous and continent women, comparing it with the activation that occurs in MVC-SP and MVC-ST.

## 2. Material and Methods

### 2.1. Study Design

The design of this study corresponds to an observational approach with a descriptive aim and a prospective, cross-sectional, temporal sequence performed in two phases: the pilot phase and the execution phase. The study was approved by the Andalusian Biomedical Research Ethics Portal (Ref: 1103-N-20), and the data were gathered in the Faculty of Health Sciences of the University of Málaga (Spain) between October 2021 and December 2021, following the principles of the Declaration of Helsinki and the rules of good clinical practice. 

### 2.2. Participants and Inclusion Criteria

The participants were recruited by opportunistic sampling among the postgraduate physiotherapy students of the University of Málaga (Spain); 37 women, who represented 34.9% of the selected population, volunteered to participate in the study without receiving any economic or academic compensation. The candidates were supplied with a comprehensive description of the aim and methods of each type of study. 

The inclusion criteria were: women aged between 18 and 40 years, nulliparous, continent (0 points in the ICIQ-SF), with a body mass index (BMI) between 18 and 30 kg/m^2^, medically healthy, capable of performing the exercises included in the FMS and a good understanding of the Spanish language. The exclusion criteria were: pain, PFM strength below 3 according to the Modified Oxford Scale, a history of urogynecological or lumbopelvic surgery and/or pelvic floor or lumbopelvic dysfunctions, acute urinary or vaginal infection, pregnancy, menopause and neurological alterations.

After obtaining their informed consent, a personal history was taken from each participant. Six women were excluded (2 participants for presenting pain during the vaginal exploration, 2 for having a score above 0 in ICIQ-SF, 1 for the muscular balance of PFM < 3 according to the Modified Oxford Scale, and 1 for acute vaginal infection). Lastly, 31 women who met the inclusion criteria participated in the study: 6 of them in the first phase of the pilot study and 25 in the second phase of the definitive intervention (Figure 1).

### 2.3. Study Variables 

The demographic data included age, height, weight, level of physical activity, use of contraceptives and smoking. To evaluate PFM strength and the level of physical activity, the Oxford Modified Scale and the International Physical Activity Questionnaire (IPAQ) (short version) [26] were used, respectively. 

In the first (pilot) phase of the study, 6 women had their PFM baseline EMG activity measured in the supine position (BASP) and standing (BAST), during MVC-SP and MVC-ST, as well as during the execution of the 7 exercises that make up the FMS: overhead squat, hurdle step, in-line lunge, shoulder mobility, rotatory stability, active straight leg raise and trunk stability push-up.

In the second phase of the study, the remaining 25 women had their PFM baseline EMG activity measured in the supine position (BASP) and standing (BAST) during MVC-SP and MVC-ST and during the execution of the FMS exercise that produced the most EMG activity in the pilot phase: trunk stability push-up (PU).

The EMG measurements were carried out using an intravaginal Periform^®^ probe (Neen Performance Health International Ltd., Sutton-in-Ashfield, UK), and the EMG signals were gathered through wireless PODs connected to the intravaginal probe, which allowed the transmission and visualisation of the data instantaneously at the central unit (PHENIX^®^ Liberty, VIVALTIS, Montpellier, France). 

Surface EMG (sEMG) amplitude data are strongly influenced by detection conditions. One solution for this problem is the normalisation of the sEMG signal to a reference value. The most common method is referred to as MVC-normalisation, referring to a maximum voluntary contraction performed both in the supine position (MVC-SP) and standing (MVC-ST). The sEMG level is then expressed as % MVC [27]. The mean of the peak values obtained in each repetition of FMS, divided by the reference value of MVC-SP and MVC-ST, determines the PFM activation level during the exercise (% MVC-SP and % MVC-ST, respectively).

### 2.4. Study Protocol

All measurements were performed to create only one set of data by two physiotherapists specialised in urogynecology. The participants did not perform any intense physical exercise within 24 h before the recordings in order to prevent possible alterations in the execution of the exercises, conducting the test in the first morning hours to minimise the potential impact of fatigue on the results. 

Prior to the recording of the measurements through EMG, the participants were asked to empty their bladders and were briefed on how to contract their PFM correctly, controlling the parasite contraction of the neighbouring muscles (adductors, glutei and superficial abdominal wall) and/or the reversal of the perineal order through vaginal bidigital palpations in the supine position in triple flexion, using nitrile gloves with a small amount of water-based hypoallergenic lubricant. The briefing they received for the contraction of PFM was “squeeze and lift my fingers as strongly as you can”, and they did not receive any feedback about the test. Once the correct contraction was ensured, they were asked to perform an MVC, and PFM strength was classified according to the Oxford Modified Scale [21]. 

Subsequently, the superficial EMG of PFM was measured, placing the intravaginal probe with water-based hypoallergenic lubricant and verifying its correct position to prevent EMG signal noise [25]. 

In the pilot phase of the study, 6 women were randomly selected among the participants. Firstly, measurements were recorded in the supine position (with the feet of the participant leaning on the stretcher): (a) BASP for 30 s, and (b) the EMG activity of MVC-SP in 3 contractions of 5 s each, leaving 1 min of rest between each repetition. Then, the same procedure was repeated with the participant standing: (a) BAST of PFM in the orthostatic position for 30 s, and (b) the EMG activity of MVC-ST in 3 contractions of 5 s each, leaving 1 min of rest between each repetition. Once these values were recorded, the maximum EMG activity of PFM was measured during the execution of the 7 exercises that make up the FMS. It is important to mention that the participants were not asked to contract PFM directly, but to perform the different tests correctly, taking the FMS guidelines as a reference [26]. A total of three repetitions of each exercise were recorded, followed by 1 min of rest between each of the exercises included in the FMS. 

In the second phase of the study, the rest of the sample (25 women) had their EMG activity of PFM measured in 3 exercises, executed in a randomised order: MVC-SP, MVC-ST and the FMS exercise that produced the most EMG activity in the previous pilot phase: PU. As in the pilot phase, the participants were not directly asked to contract their PFM but to perform the PU exercise correctly, with the same references and repetitions as in the previous phase (Figure 2).

### 2.5. Statistical Analysis

The repeated measures ANOVA test and Friedman’s test were used to identify differences in the measurements of the three exercises. To explore the correlation between numerical variables, Pearson’s correlation coefficient was used. All the statistical analyses were carried out using the statistical software Jamovi v1.6 (The Jamovi Project. Sydney, Australia) [28,29,30]. 

## 3. Results

The demographic data of all participants are shown in Table 1. The average age of the participants was 26 years (*SD* = 3.2), the mean BMI was 22.3 kg/m^2^ (*SD* = 3.3) and the average muscle balance in the exploration was 4.3 (*SD* = 0.7).

All the tests and exercises of the FMS performed by the six participants in the pilot phase showed a value under 100% MVC (both SP and ST) except PU, which obtained a mean value of 101.3 μv (*SD* = 54.5), that is, 112% MVC (*SD* = 37.6); therefore, this exercise was selected for the next phase of the study.

In the second phase, the remaining 25 women continued with the study protocol, performing only PU from the FMS. The results are presented in Table 2.

We used Shapiro–Wilk tests and Q-Q graphs and found no evidence against normality in any of the main variables: MVC-SP, MVC-ST, PU, PU/MVC-SP and PU/MVC-ST (*p* > 0.274 in all cases).

Neither the parametric ANOVA test of repeated measures (*p =* 0.365) nor Friedman’s non-parametric test (*p =* 0.087) revealed differences between the three exercises (MVC-SP, MVC-ST and PU). The data did not deviate from normality.

A 95% confidence interval for PU/MVC-SP and PU/MVC-ST ratios with respect to MVC were [99–136] and [94–129], respectively.

The PU exercise does not appear to be inferior in terms of MSP activation with respect to MVC-SP and MVC-ST.

Among the biometric variables, a significant correlation was only found for BMIs with an MVC-SP/PU ratio of r = 0.473, *p =* 0.017 and with an MVC-ST/PU ratio of r = 0.426, *p =* 0.034.

## 4. Discussion

The present study has analysed and compared the EMG activity of PFM during the PU exercise of the FMS in 25 nulliparous and continent women with maximum voluntary contractions of PFM in the supine position and standing (MVC-SP and MVC-ST, respectively). The results show no evidence of significant differences between the three analysed exercises. Despite the fact that the participants were not asked to voluntarily activate their PFM during the execution of the PU exercise, the latter presented better EMG values than MVC-SP and MVC-ST.

With the aim of justifying the influence of exercise on the integrity of the pelvic floor, two exercise modalities have been reported to increase IAP to a greater extent than others, and thus they may have a negative effect on the pelvic floor: extenuating strength training (e.g., weight lifting), and high-impact activities, such as jumping and running [24]. The habitual practice of the PU exercise, often included in current training and physical activity programmes (similar to the typical “planking”), is not contemplated within such group; therefore, its recommendation in healthy women should not be banished. 

The PU exercise is defined as an exercise for the training and valuation of lumbopelvic stability. Thus, it is logical that, during its correct execution, given the direct activation of the transverse abdominal muscle (whose fibres are directly connected to those of the transverse perineal muscle) [31], PFM participates by playing one of its main functions: lumbopelvic stabilisation [32,33,34]. During its execution, a pattern of global trunk stability is required, connecting the lower and upper parts of the body, thereby producing an unconscious and/or reflex co-contraction of the PFM [35], which could explain the EMG activation achieved. Such a result, in our opinion, creates an indirect training effect that contributes to reducing the levator ani hiatal area by causing the hypertrophy and shortening of the surrounding muscles, thus lifting the pelvic floor and the internal organs to a higher pelvic location. Theoretically, such morphological changes could reduce the risk of urinary incontinence, faecal incontinence and pelvic organ prolapse [24].

Moreover, the correlation between % MVC in the supine and standing positions and BMI could be due to the generation of a greater increase of IAP at higher BMI values, which would be in line with previous studies [36,37,38]. This increase in IAP would generate a greater PFM activity, with the aim of maintaining pelvic visceral statics and guaranteeing continence. 

In any case, the IAP generated and managed during the exercise is still a controversial topic. Different studies clearly show that the maximum values of IAP have a wider range among women who perform the same standardised activity [39], and the maximum IAPs vary among studies for the same activity, partly due to the instruments used to measure IAP, the way in which maximum IAP is created, and the differences between populations [24]. In the same line, an interesting study questioned the idea that “safe” exercises for the pelvic floor generate lower IAPs than conventional exercises since no differences were found in the IAP values between the recommended and ill-advised versions of half of the exercises [21]. Another study highlights that the activities that are generally restricted after surgery may generate lower IAP values than non-restricted activities (e.g., the maximum average IAP was higher when the participant stood up from a chair than climbing stairs, doing crunches and lifting weights) [25].

However, EMGs do not provide information about the real effect of IAP increase on the statics and displacement of intra-pelvic structures. In this sense, the possible development of imaging techniques that allow observing, in real-time, the displacement of the pelvic organs in situations of IAP increase or muscular activation is in the hands of bioengineering. 

It is fundamental to consider the fact that the EMG values in the three types of contraction tests were similar in a young sample capable of performing a high activation of their PFM. The reproduction of this study in a larger and more representative sample, as well as the comparison of the results with women in whom the pelvic floor function can be affected by age, high BMI values, vaginal births and the habitual practice of risk exercises (extenuating strength training, running or jumping), will be an interesting aspect to explore in future studies [40]. 

It is widely accepted that the effectiveness of exercises for PFM depends on the adherence of the patient to them, essentially derived from self-efficacy, finding a positive correlation between the two factors (effectiveness–self-efficacy) [2,41,42]. In healthy and nulliparous women, this could be a good strategy to prevent pelvic floor dysfunction (especially urinary incontinence), encouraging them to practise this type of functional exercise in sports centres, where there may be greater self-efficacy due to the belief about the benefits derived from it (general and indirect benefits on the pelvic floor), as well as a better capacity to execute them and the social support received. This could generate behavioural changes in the attitude of women, which will make them participate more actively and with greater commitment in their preventive treatment, minimising the dropout rates in PFM training derived from the responsibility of performing the Kegel exercises on their own without receiving any feedback. 

In any case, due to the existing knowledge gaps and the lack of further research, it is not possible to draw solid conclusions about the effect of physical activity on the rate of pelvic floor dysfunctions, highlighting the need for more high-quality studies that clarify this aspect [24].

### Limitations of the Study

Although EMG is a method that allows us to evaluate muscle activity in an objective and reliable manner, it is necessary to mention the cross-talk phenomenon. This is defined as the recorded EMG activity that comes from the neighbouring muscles rather than exclusively from the study target muscles [43,44], which could alter the final results and lead to erroneous interpretations. During functional exercises, large muscle groups are activated, such as the gluteus maximus, and although its influence on the EMG result has not been demonstrated yet, this factor should be taken into account. 

It is also important to consider motion artifacts, which can distort the EMG signal. This study used a wireless system and a Periform^®^ probe, due to its pear shape and the longitudinal placement of the metallic plates, with the aim of reducing such phenomenon, although it is necessary to develop new intravaginal EMG probe designs in order to improve EMG measurements in research and in clinical practice [25]. 

Moreover, three repetitions of each exercise were used, which does not guarantee a good activation of the musculature with the increase in repetitions. That is, further information is required about the fatigability of PFM, both in a session and in the long term between sessions. 

The data provided by this pilot study “tend” to explain that PU can activate PFM muscles; however, it is necessary to carry out more robust methodological and statistical research with larger samples to advance such statements.

In addition to expanding the sample, it is also necessary to include other symptomatic and/or older populations in order to continue defining new prevention and treatment protocols for perineal health and new types of training that increase the adherence of users. 

## 5. Conclusions

In the nulliparous and continent women of our sample, there was no evidence of significant differences in the EMG activation of PFM between the three exercises analysed: MVC-SP, MVC-ST and PU. The results show better EMG values in PFM for the functional exercise of PU. This functional exercise, which is often included in current functional training programmes, may be, on one hand, training the PFM indirectly and, on the other hand, promoting behavioural changes in the attitude of women that could encourage them to participate more actively and with greater commitment in their preventive treatment, thus minimising the dropout rates that occur with classic PFM training. 

## Figures and Tables

**Figure 1 healthcare-11-00561-f001:**
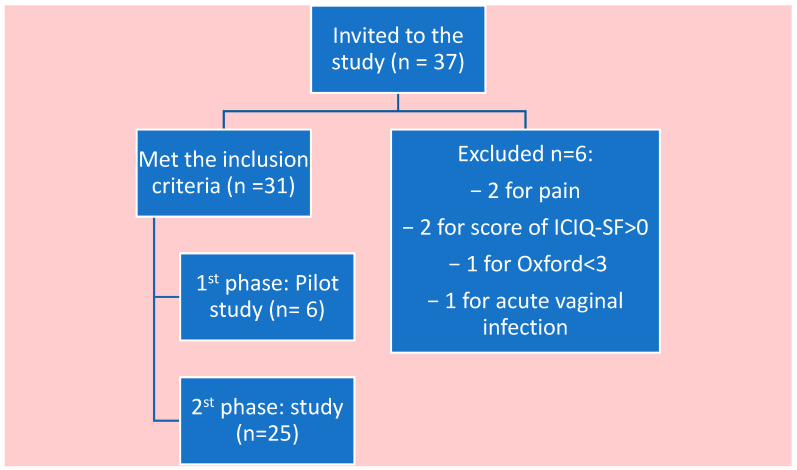
Flowchart of the participants of the study.

**Figure 2 healthcare-11-00561-f002:**
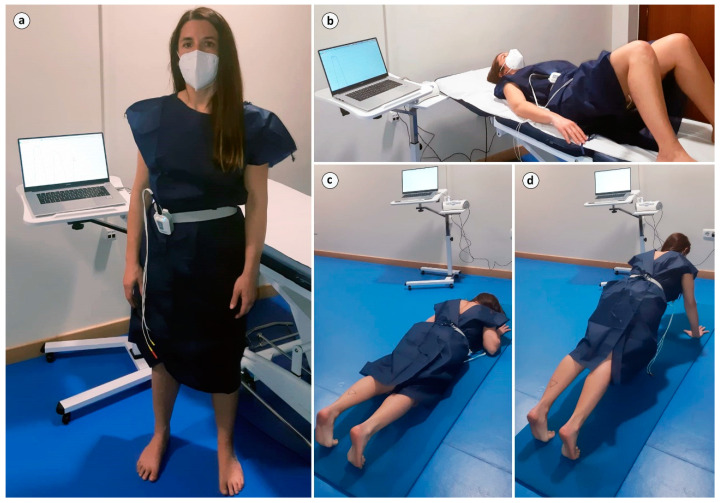
EMG activity measured in (**a**) the supine position (BASP), (**b**) standing (BAST) and (**c**,**d**) trunk stability push-up (PU).

**Table 1 healthcare-11-00561-t001:** Demographic data of the participants.

Demographic Data (n = 31)		
	Mean (SD)	Range
Age	25.9 (3.0)	23.0–38.0
BMI	22.3 (3.1)	18.3–29.7
Muscular balance (Oxford)	4.3 (0.7)	(3–5)
Hormonal contraceptives (yes, no (%))	2 (6.4%)	
Smoking (yes, no (%))	4 (12.9%)	
Moderate physical activity level	20 (64.5%)	

**Table 2 healthcare-11-00561-t002:** Electromyographic values (μV and % MVC) of baseline activity in the supine position (BASP) and standing (BAST), and maximum voluntary contraction in the supine position (MVC-SP), standing (MVC-ST) and during the push-up (PU) exercise.

EMG Activity of PFM (μV) (n = 25)		
Measures	Mean (SD)	Range
BASP	4.0 (3.2)	1.0–16.0
BAST	6.8 (3.7)	2.0–17.0
MVC-SP	39.2 (10.4)	21.7–57.3
MVC-ST	37.5 (10.4)	18.0–57.3
PU	40.7 (10.2)	17.3–55.7
PU/MVC-SP Ratio (%)	112 (42.6)	33.8–225
PU/MVC-ST Ratio (%)	118 (45.0)	41.9–236

Abbreviations: baseline activity in the supine position (BASP), baseline activity standing (BAST), maximum voluntary contraction in the supine position (MVC-SP), maximum voluntary contraction standing (MVC-ST), push-up (PU), the ratio of push-up/maximum voluntary contraction in the supine position (Ratio PU/MVC-SP) and the ratio of push-up/maximum voluntary contraction standing (Ratio PU/MVC-ST).

## Data Availability

Data sharing not applicable.

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
