# Peer review of "Functional Exercise Versus Specific Pelvic Floor Exercise: Observational Pilot Study in Female University Students"

_healthcare, 2023, doi:10.3390/healthcare11040561_

Round 1
Reviewer 1 Report
Dear Authors,
congratulations for your work.
I think it's an interest and understimate topic, useful in partucular to select what kind of muscolar activity shoul be preferred to potentiate and should not be leave behind for young continent women
I have no great comments or suggestions for what concern this paper, at least for the methods and the topic. Maybe you should consider to enrich a bit your results.
Greetings again,.
Author Response
We very much appreciate your comments.

Author Response
Consulte el archivo adjunto

Round 2
Reviewer 2 Report
I suggest those minor corrections and a review of the english and then I could be published:
In the abstract: "Anova test, Friedman’s test and Pearson’s correlation coefficient was used. Perhaps you should detail just a little bit, something like: Anova, Friedman’s and Pearson’s tests where used to compare means and observe correlations.
Introduction paragraph #3 (b) General (should be :general)
Statistical analysis
Anova test (which kind of Anova?) and Friedman’s test were used to identify differences (between the means? )in the measurements of the three exercises.
